# EMT Dynamics in Lymph Node Metastasis of Oral Squamous Cell Carcinoma

**DOI:** 10.3390/cancers16061185

**Published:** 2024-03-18

**Authors:** Yasmine Ghantous, Shiraz Mozalbat, Aysar Nashef, Murad Abdol-Elraziq, Shiran Sudri, Shareef Araidy, Hagar Tadmor, Imad Abu El-naaj

**Affiliations:** 1Department of Oral and Maxillofacial Surgery, Tzafon Medical Center, Faculty of Medicine, Bar Ilan University, Ramat Gan 5290002, Israelsudri.shiran@gmail.com (S.S.); shareef_ara@yahoo.com (S.A.); iabu@poria.health.gov.il (I.A.E.-n.); 2Molecular Biology of Oral Cancer Laboratory, Tzafon Medical Center, Azrieli Faculty of Medicine, Bar-Ilan University, Safed 1311502, Israel

**Keywords:** oral carcinoma, EMT, lymph node, N-cadherin

## Abstract

**Simple Summary:**

Epithelial–mesenchymal transition (EMT) enables tumor cell invasion and metastasis. Many studies have demonstrated the critical role of EMT in lymph node metastasis in oral squamous cell carcinoma (OSCC). This study involved the genetic profiling analysis of 159 primary OSCCs TCGA samples using bioinformatic tools. The analysis focused on the expression of EMT markers, including cadherin switch genes (N-CDH, E-CDH), TGF-β/SMAD pathway genes, SNAIL, and keratin genes. The samples were categorized into advanced stage (stage III–IV) and early stage (stage I–II). Developing therapies targeting regulators such as N-cadherin may prevent metastasis and improve outcomes. Further research is warranted to elucidate EMT signaling in OSCC progression.

**Abstract:**

Background: Epithelial–mesenchymal transition (EMT) enables tumor cell invasion and metastasis. Many studies have demonstrated the critical role of EMT in lymph node metastasis in oral squamous cell carcinoma (OSCC). During EMT, epithelial cancer cells lose intercellular adhesion and apical–basal polarity and acquire mesenchymal properties such as motility and invasiveness. A significant feature of EMT is cadherin switching, involving the downregulation of E-cadherin and upregulation of N-cadherin. The TGF-β/SMAD pathway can also induce EMT. We aimed to evaluate EMT markers as predictors of lymph node metastasis in OSCC. Methods: We performed genetic profiling of 159 primary OSCCs from TCGA and analyzed the expression of EMT markers, including cadherin switch genes (CDH1, CDH2), and TGF-β/SMAD pathway genes. Samples were divided into advanced (stage III–IV) and early (stage I–II) stage groups. Differential expression analysis was performed, as well as an independent validation study containing fresh OSCC samples. Results: TGF-β/SMAD pathway genes such as SMAD6 were upregulated in advanced stage tumors. N-cadherin and SNAIL2 were overexpressed in node-positive tumors. Keratins were downregulated in these groups. Conclusion: Our findings demonstrate that EMT marker expression correlates with lymph node metastasis in OSCC. Developing therapies targeting regulators such as N-cadherin may prevent metastasis and improve outcomes.

## 1. Introduction

Oral squamous cell carcinoma (OSCC) is a malignant epithelial tumor arising from the mucosal surfaces of the oral cavity. Despite advances in treatment, OSCC continues to have poor survival rates, with a 5-year survival rate of approximately 50% [1]. Cervical lymph node metastasis is the most significant prognostic factor, mandating treatment intensification and leading to poorer outcomes [2,3].

The development of lymph node metastasis requires that tumor cells detach from the primary site, migrate through the lymphatic system, and establish new tumors in the lymph nodes [4]. This process involves a phenotypic transition known as epithelial–mesenchymal transition (EMT). During EMT, epithelial tumor cells lose intercellular adhesion properties and apical–basal polarity and acquire increased motility and invasiveness [5,6]. A critical molecular alteration in EMT is cadherin switching, which is the downregulation of E-cadherin and upregulation of N-cadherin. E-cadherin is an adhesion molecule that maintains intercellular contacts in epithelial tissues. The loss of E-cadherin disrupts adherens junctions, releasing β-catenin into the cytoplasm. This activates signaling pathways involved in EMT, such as TGF-β/SMAD [7,8]. The concomitant increase in N-cadherin further augments cell motility and invasion [9].

Other EMT-related pathways include upregulating transcriptional factors such as SNAIL, ZEB1, and TWIST, which repress E-cadherin expression [10]. Transforming growth factor beta (TGF-β) signaling can induce EMT through SMAD-dependent and SMAD-independent mechanisms [11].

In the context of OSCC, EMT is influenced by several microenvironmental factors, such as cytokines, enzymes, growth factors, and proteins responsible for cell–cell interactions [9]. The cross-talk between the tumor cells and their stroma is paramount in promoting the EMT process, tumor invasion, and metastasis [8,9]. Apart from the well-known EMT markers, such as E-cadherin, SMAD, SNAIL, and TWIST, in OSCC, several other features are associated with tumor progression and lymph node metastasis. Matrix metalloproteinases (MMPs), fibronectin, and vimentin provide valuable insight into the EMT process in OSCC, while their expression may determine the aggressiveness of the tumor and metastasis potential [7,10,11].

Overall, the induction of EMT allows OSCC cells to acquire a mesenchymal phenotype permissive for migration and metastasis [12,13]. Elucidating the molecular underpinnings of EMT may reveal therapeutic targets to halt lymph node spread.

In this study, we aimed to evaluate the significance of EMT markers, particularly cadherin switching and TGF-β/SMAD signaling, in predicting lymph node metastasis in OSCC. We performed genetic profiling of primary OSCCs to examine the expression of EMT-related genes concerning cervical node status.

## 2. Materials and Methods

Molecular data sets of 528 head and neck carcinoma patients were obtained from the TCGA data portal (https://cancergenome.nih.gov/, accessed on 1 January 2023) and the Genome Data Analysis Center (GDCA). Genomic processing of the molecular data sets was performed using cBioPortal for Cancer Genomics analysis (http://www.cbioportal.org/, accessed on 1 January 2023).

The molecular datasets included genetic analysis based on whole-genome sequencing. The HPV status was defined using an empiric definition of >1000 mapped RNA-seq reads, primarily aligning to viral genes E6 and E7 [14]. The HPV status by mapping DNA-seq reads was concordant with the genomic, sequencing, and molecular data and indicated that 36 tumors were HPV(+) and 243 were HPV-negative (Appendix A). To eliminate unnecessary molecular or genetic diversity, only HPV-negative and pathologically proven oral cavity tumors were included in this study (*n* = 159), and 72 cases of larynx carcinoma, 2 cases of hypopharyngeal carcinoma, and 10 cases of tonsils carcinoma were excluded (Appendix A). The selected genes (MET genes) that were analyzed were VIM, SNAIL1, SNAIL2 and TWIST, keratins genes (51 genes), SMAD genes, (by mSIGdb: http://www.gsea-msigdb.org/gsea/msigdb/, accessed on 1 January 2023), cadherin genes (CDH1, CDH2), and TGF pathway genes (84 genes); overall, 413 genes were analyzed.

Statistical and bioinformatic analyses were conducted to elucidate the genetic factors associated with lymph node metastasis development and other clinicopathological features. Two primary analyses were performed. The first analysis aimed to identify differentially expressed genes between the groups using the Dseq2 analysis for 413 genes. The traditional Bonferroni test was applied to adjust the *p*-value and reduce the false positive rate, resulting in an adjusted *p*-value of 0.1. Subsequently, significantly differentially expressed genes were highlighted, with a positive fold change indicating increased expression in the advanced stage group and a negative fold change indicating increased expression in the early stage group (Figure 1).

The second part of the analysis and the main focus of the current study sought to identify differentially expressed genes among the studied genes concerning various clinical and pathological parameters such as age, gender, pTNM staging (divided into two main groups: early-stage, which included samples that were staged pathologically as stage I and II according to the AJCC 8th edition, and advanced-stage, which included tumors that were pathologically staged as stage III and IV); pT stage, pN stage, recurrence, and primary tumor site. This was statistically evaluated using either the Kruskal–Wallis or Wilcoxon test, depending on the number of groups within each feature, with a *p*-value set at 0.05. Overall survival (OS) and recurrence-free survival (RFS) were estimated from the clinically available data using Kaplan–Meier analysis. Follow-up time was defined as the time that passed from the date of the initial diagnosis as seen on the pathological report of the biopsy until either the date of death or the last clinical follow-up as recorded in the files.

### Study Validation

The independent validation cohort (designated as the immunohistochemistry staining cohort) contained 28 unstudied OSCC tumors from a native surgical resection (without a history of chemotherapy or radiation).

Tissues from the tumor samples were analyzed for E-cadherin and N-cadherin presence. Staining of the formalin-fixed, paraffin-embedded 5-micron tissue array section was performed. First, slides were deparaffinized with xylene and rehydrated, and endogenous peroxidase was quenched with 3% hydrogen peroxide in methanol. Slides were then subjected to antigen retrieval by boiling in citrate buffer at pH 6, blocked with 10% normal goat serum, and incubated with the primary antibody for E-cadherin (1:100, ab40772, Abcam, Cambridge, UK) and N-cadherin (1:100, ab76011, Abcam). Following additional washes, color was developed using the AEC reagent (Sigma, St. Louis, MO, USA), and sections were counterstained with hematoxylin and mounted. Independent investigators scored slides. Tissue MicroArrays^®^ (Abcam, Cambridge, UK) was used to enable a fast, accurate, standardized method for screening cancer biopsies.

## 3. Results

Data on gene expression were available in TCGA for 159 patients (OSCC patients, HPV negative).

Clinical and pathological data of the entire cohort are summarized in Table 1. The mean age at diagnosis was 62 ± 13 years, with a male-to-female ratio of ~2:1. Eighty-two (51%) patients had a history of tobacco exposure (with an average of 47 pack-years), and 101 (63%) patients reported alcohol consumption. The mean follow-up for the entire cohort was 26 months.

The most common primary tumor site was the oral tongue (44%), followed by the floor of the mouth (15%), with the remainder distributed between the buccal mucosa, hard palate, and alveolar ridges.

Lymph node dissection (selective and radical) was performed in 137 (86%) patients. Seventy (44%) patients had lymph node metastasis, as seen on histopathology, with an average of 2 positive lymph nodes for each patient.

The study cohort was divided into two groups based on pathological staging. Advanced-stage samples included pathologically staged patients for stages III and IV (112 patients), and early-stage samples included stages I and II (47 patients).

### 3.1. Cross-Tab Analysis: Advanced Stage vs. Early Stage (Table 1)

According to the TNM staging (8th edition), the study population was divided into two subgroups, the early-stage group and the advanced-stage group. The advanced-stage group included 112 patients: 41 patients were diagnosed with stage III disease, while the remaining 71 had stage IV disease. The male-to-female ratio in this group was 78/34, and the mean age at diagnosis was 61 ± 12 years. Seventy-seven (69%) patients in this group also reported substantial tobacco exposure (average 28 packs/year); this finding was significantly higher than that in the early-stage group (*p* value = 0.001).

Most of the index primary tumors in the advanced-stage group, similar to the study cohort as a whole, were in the oral tongue (46 patients, 41%), followed by the floor of the mouth (21 patients, 19%). In the early-stage group, the oral tongue was also the primary tumor site (24 patients, 51%), while the floor of the mouth comprised only 5% of the primary tumors (*p*-value = 0.04).

Although recurrence is not the only factor determining the survival of OSCC patients, the patients’ survival status was studied as a function of the recurrence status at the last follow-up visit. Disease-free survival was significantly lower among patients in the advanced-stage group (mean: 19 months versus 36 months, *p* value = 0.01).

### 3.2. Genetic Profiling Analysis

We performed genetic profiling of 159 primary tumors for the plotting of the selected MET genes (VIM, SNAIL1, SNALI2, and TWIST1), keratin genes, which included 51 genes (KRT1, KRT10, KRT13, KRT14, KRT15, etc.); cadherin switching pathway genes, i.e., downregulation of E-cadherin and upregulation of N-cadherin in each group (comparative RNA analysis). We also analyzed TGF pathway genes, such as BAMBI, CREBBP, EP300, E2F4, and SMAD gene expression.

For pathological TNM staging, the genetic analysis identified a set of TGF signaling genes to be differentially expressed in the different stage groups. The gene CREBBP was significantly upregulated in stage IVA and stage IVB patients, and the CREBBP gene plays a multifaceted role in cancer development and progression. Its involvement in transcriptional regulation, chromatin remodeling, and signaling pathways is significant for cancer progression and tumorigenesis.

On the other hand, SMAD6, CHRD, and PITX2 were found to be upregulated in stage I patients. SMAD6 is known for its tumor suppressor role; its upregulation in cancerous tissue predicts a better prognosis. CHRD is involved in embryonic development and cell signaling pathways. Its role in cancer progression is not yet known; however, wild-type CHRD may have a positive effect due to its regulatory role in the signaling pathway. Although the expression of PITX2 was higher in the stage I group, it was overexpressed in stage III and stage IVA patients, which may indicate a multifactorial role in tumor progression. BMP4 was slightly upregulated in stage IVA patients, while the EP300 gene was remarkably upregulated as the tumor stage advanced, especially in stage III patients (Figure 2A,B).

For pathological N staging, all the investigated gene groups were found to have a role in determining the N stage. In the TGF group, the genes ACVR1, PPP2R1B, GDF7, RBX1, and MAPK3 were upregulated in positive N pathological staging, and SNAIL2, which is known for promoting migration in specific cells during the development process, was also significantly upregulated in the N positive group, especially in the N2 group. On the other hand, keratin genes were upregulated in pathological N negative patients. The KRT80 gene was significantly upregulated in N negative patients, while KRT78 was upregulated in both N negative and pN1 patients. The CDH2 gene is a neuronal cadherin, also known as N-cadherin, associated with the cadherin switching pathway, and was significantly upregulated in the pathologically positive N stage. This gene is the critical gene in the formerly mentioned pathway, and its upregulation indicates the presence of the EMT process (Figure 3). Also, it is most important to demonstrate that the genes SNAIL2 and CDH2 were remarkably overexpressed in the N2 group, which marks the ability of the tumor cells to migrate to several lymph nodes.

Keratin and TGF genes were also significantly overexpressed in advanced diseases. EP300 was upregulated in the T4 and T1 groups; however, its expression was remarkably higher in the T4A groups. On the other hand, the genes TNF and KRT76 were significantly upregulated and overexpressed in the T1 group, which indicates their role as tumor suppressor genes, inhibiting tumor progression and invasion. KRT18 and AMHR2 were upregulated in the advanced T stage, as seen in Figure 3; their expression increased as the T advanced, demonstrating those genes’ role in tumorigenesis and progression, especially in cancerous cell proliferation. RPS6KB1 was slightly upregulated in group T4A compared to T1, and KRT86 was upregulated in T1 groups, especially compared to the T4B group (Figure 4).

### 3.3. Kaplan–Meier Survival Testing

Kaplan–Meier survival tests were performed to evaluate advanced and early tumor stages to metadata information. All the selected genes and cadherin genes were evaluated (MET genes).

We found that tumor samples that overexpressed the MET genes (CDH2 and TGF genes) had a decreased survival probability in relation to the overall mean expression of those genes (Figure 5).

### 3.4. Independent Validation Using IHC

To validate the genetic analysis results, we assessed the protein expression level of the primary key genes, E-cadherin and N-cadherin, using immunohistochemistry staining of E-cadherin and N-cadherin proteins in an independent cohort of 28 primary OSCC tissue samples; of this cohort, 13 samples were staged pathologically as N0, and the other 15 patients were staged pathologically as N positive (Table 2). Before proceeding with the immunohistochemical analysis, an independent head and neck pathologist reviewed two sections of each sample (including tissue margins) to verify the presence of oral squamous cell carcinoma.

The relative expression mode of each protein was evaluated using Image J software (V 1.8.0). Notably, the average % T-area expression of E-cadherin in N negative samples was significantly higher (*p* value = 0.019) than the overall average of E-cadherin expression of the whole cohort and in the N positive group (25.919, 23.091, and 20.263 respectively). In the same manner, the average % T-area expression of N-cadherin was significantly higher in the N positive group (0.446) when compared to the N negative group (0.171) *p* value = 0.037 (Figure 6).

## 4. Discussion

Epithelial–mesenchymal transition (EMT) is a well-known mechanism in tumor advancement and progression. In this biological process, epithelial cells, characterized by tight cell-to-cell adhesion and basal polarity, lose these properties and acquire weak cell-to-cell interactions, thus enabling increased migration and transition ability [4]. EMT markers have been known to promote metastasis and invasion in various types of cancers, such as breast cancer, lung cancer, colorectal cancer, and oral cancer [3,6,9,15].

In this study, we investigated the significance of EMT markers in predicting lymph node metastasis and promoting several histopathological features in OSCC patients.

The differential expression of TGF signaling genes, including ACVR1, PPP2R1B, GDF7, RBX1, and MAPK3, has been linked to their role in tumor progression and metastasis in the current study, as they were overexpressed in advanced disease in terms of tumor size and metastatic cervical lymph nodes samples. Several whole-transcriptome chips and subsequent tissue microarray analyses have demonstrated that the MAPK3 signaling pathway is associated with lymph node metastasis in OSCC [14,16,17], while high levels of MAPK-related proteins were associated with advanced tumor stage and lymph node metastasis [14]. In addition, the expression of TGF-β results in metabolic dysfunction and promotes EMT, which may lead to fibrosis and cancer cell transition [12,13]. Additionally, activated TGF-β stimulates different downstream signaling pathways; as a result, other kinases and signaling pathways, which generally act as expression-regulating proteins, become dysfunctional; thus, uncontrolled proliferation and cell migration occur [15,16,18]. One of the major EMT biological processes that was extensively studied in this study is the switch from E-cadherin to N-cadherin, which initiates several processes, including mediating cell adhesion weakness by degrading intracellular contacts and cell-to-cell adhesions, degrades intracellular connections, promotes cancer cells to detach from their primary environment, and enables them to invade the adjacent lymphatic system and adjacent tissue. Concurrently, the upregulation of N-cadherin promotes cell motility and increases resistance to apoptosis, facilitating lymphatic colonization and metastasis [2,4,7].

In the current genetic analysis, N-cadherin, also known as CDH2, was significantly upregulated in N positive patients samples’, especially in the N2 group, and downregulated in the N0 and N1 groups. Moreover, in the validation study, N-cadherin was predominantly expressed in the N positive group samples, while E-cadherin was extensively immune-stained in the N negative samples.

Angadi et al. also reported the same noticeable change of staining of N/E-cadherin, and concluded that this shift was essential for OSCC cells to enter the underlying connective tissue and during lymph node metastasis [19]. Kaur et al. and Gonzalez-Moles et al. demonstrated the crucial role of E-cadherin’s activity in its interaction with catenin, a cytoplasmic adaptor protein. This interaction binds the E-cadherin terminal tail to the actin cytoskeleton, facilitating cellular adhesion. The absence of this cadherin–catenin complex increases the likelihood of a cytoplasmic shift, leading to loss of cellular adhesion [20,21]. Consequently, the shift to N-cadherin in the validation study is linked to heightened malignant invasiveness, high histological grade, and lymph node metastasis, as it was mainly expressed in advanced-stage patients. Numerous studies have highlighted that disrupted cell–cell adhesion plays a crucial role in promoting cell motility and invasion [22,23,24,25], which are essential for the establishment of new metastasis. Intriguingly, despite displaying an EMT-like phenotype, LNM specimens consistently exhibited normal E-cadherin levels. This suggests that tumor cells within LNM may employ multiple survival strategies, potentially modulating cell division rates while upholding E-cadherin expression, although displaying characteristics associated with EMT core regulators such as TWIST and SNAIL [26,27,28].

SNAIL2, known for promoting migration in specific cells in the development process, was also significantly upregulated in the N positive group in the current study. SNAIL1 and SNAIL2 are master genes in regulating the expression and action of E-cadherin [29]. Several OSCC models have shown that the expression of SNAIL can successfully transfer epithelial cells into a fibroblast-like appearance, which includes vimentin filaments, E-cadherin/N-cadherin switching, and an almost complete lack of cell-to-cell adherence, and hemidesmosomes [30,31,32,33]. SNAIL2 also acts as a repressor of E-cadherin transcription since it binds to the E-box in the promoter region of the E-cadherin gene and inhibits its transcription; it also induces the activity of histone deacetylase genes, which remove the acetyl group from the histone protein, consequently resulting in further prevention of the E-cadherin gene [33]. This dual suppression ability resulted in tumor metastasis in several in vivo studies [34,35,36].

Another factor that may induce EMT in OSCC is the oral microbiota. The oral cavity harbors a distinctive microbiota comprising over 1000 microbial species, including commensals and opportunistic fungi, bacteria, and viruses. Alterations in this microbial community lead to dysbiosis, exacerbating the onset of oral diseases [37,38]. Species such as *Fusobacterium nucleatum*, *Porphyromonas gingivalis*, and *Prevotella* sp. have been linked to oral squamous cell carcinoma (OSCC). Prolonged infection of *P. gingivalis* promotes migratory and invasive properties in epithelial cells, inducing the expression of matrix metalloproteinase and epithelial–mesenchymal transition (EMT) transcription factors such as SNAIL, SLUG, and ZEB1. *P. gingivalis* infection activates signaling cascade pathways such as ERK1/2-Ets1, p38/HSP27, and PAR2/NF-κB to regulate the expression of ProMMP9. Additionally, the gingipain protease produced by *P. gingivalis* induces the activation of ProMMP9. Phospho-GSK3β, an EMT marker, increases significantly in oral primary epithelial cells during *P. gingivalis* infection, along with increased expression of EMT transcription factors and decreased expression of adhesion molecules. Long-term infection exacerbates the migration of *P. gingivalis*-infected epithelial cells and its invasion to neighboring tissue and the lymphatic system [39,40]. In addition, *P. gingivalis* LPS exacerbates MMP9 secretion by human monocyte-derived dendritic cells, as demonstrated by various dosages inducing MMP9 secretion. Live *P. gingivalis* strains induce MMP9 and TIMP-1 production, with major and minor fimbriae playing distinct roles in binding and invasion of dendritic cells. Different strains of *P. gingivalis* influence elevated MMP9 secretion, reflected in upregulated MMP/TIMP-1 ratios. Similarly, *Fusobacterium nucleatum* increases MMP13 production in epithelial cells, activating MAP kinase p38 and promoting cellular migration through Etk/Bmx, S6 kinase p70, and RhoA kinase. These pathways are known to regulate cytoskeleton organization, focal adhesions, and cellular migration [39]. *Fusobacterium nucleatum*, similarly to *P. gingivalis*, can also induce EMT by utilizing adhesins such as FadA, Fap2, and outer membrane protein A to bind and invade epithelial cells. FadA specifically binds to E-cadherin, promoting attachment and invasion, ultimately activating β-catenin signaling and leading to the upregulation of Wnt genes, transcription factors, N-cadherin and other oncogenes including myc and cyclin D1. This process involves phosphorylation and internalization of E-cadherin, affecting the phosphorylation and nuclear translocation of β-catenin, which eventually leads to degradation of the basement membrane, and migration of the epithelial cells [41,42].

Another interesting finding in the current study was that patients in the advanced group reported a significantly higher tobacco consumption rate compared to the early-stage group. Several studies have demonstrated that nicotine significantly impacts tumor cells’ epithelial–mesenchymal transition (EMT) [43,44,45]. Nicotine, the primary component of tobacco, can induce oxidative stress in tumor cells. This oxidative stress can trigger signaling pathways that promote EMT. As a result, tumor cells treated with nicotine demonstrate increased expression of mesenchymal markers such as vimentin, SNAIL, and PrX1 proteins while showing decreased levels of epithelial markers such as E-cadherin [38,39,40,41,42,43]. The upregulation of mesenchymal markers and downregulation of epithelial markers in response to nicotine exposure suggest that nicotine may play a role in promoting the transition of tumor cells toward a more aggressive, invasive phenotype. This transition is a hallmark of EMT and is associated with enhanced metastatic potential and resistance to cancer treatments [45].

The clinical implications of these findings are substantial, as they highlight the potential of genetic analysis in informing prognosis, predicting metastatic risk, and guiding treatment strategies for oral cancer patients. By identifying stage-specific gene expression patterns, clinicians can gain valuable insights into the molecular landscape of the disease and tailor treatment plans to address the varying biological behaviors associated with different pathological stages. This personalized approach holds promise for improving patient outcomes and addressing the challenges posed by tumor heterogeneity and metastatic potential in oral cancer.

## 5. Conclusions

To conclude, understanding the intricate interplay between EMT, N-cadherin switching, SMADs, SNAILs, and lymph node metastasis is crucial in unraveling the molecular mechanisms underlying oral cancer progression. It appears that those genes are responsible for the EMT progression, and their expression enables tumor cells to migrate and invade the cervical lymph nodes. Further research is warranted to fully elucidate the signaling pathways and molecular players involved in these processes. Targeting N-cadherin switching holds excellent potential for developing effective therapeutic interventions to prevent or treat lymph node metastasis, ultimately improving oral cancer patients’ survival rates and quality of life.

## Figures and Tables

**Figure 1 cancers-16-01185-f001:**
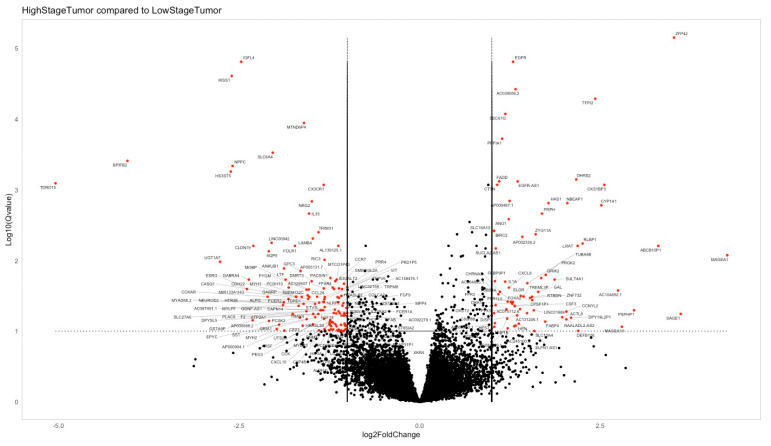
Group comparison for differentially expressed genes. Positive fold change denotes increased expression in the advanced stage group, while a negative fold change signifies increased expression in the early stage group. The genes marked in red are significantly differentially expressed (Bonferroni adj. *p*-Value = 0.1).

**Figure 2 cancers-16-01185-f002:**
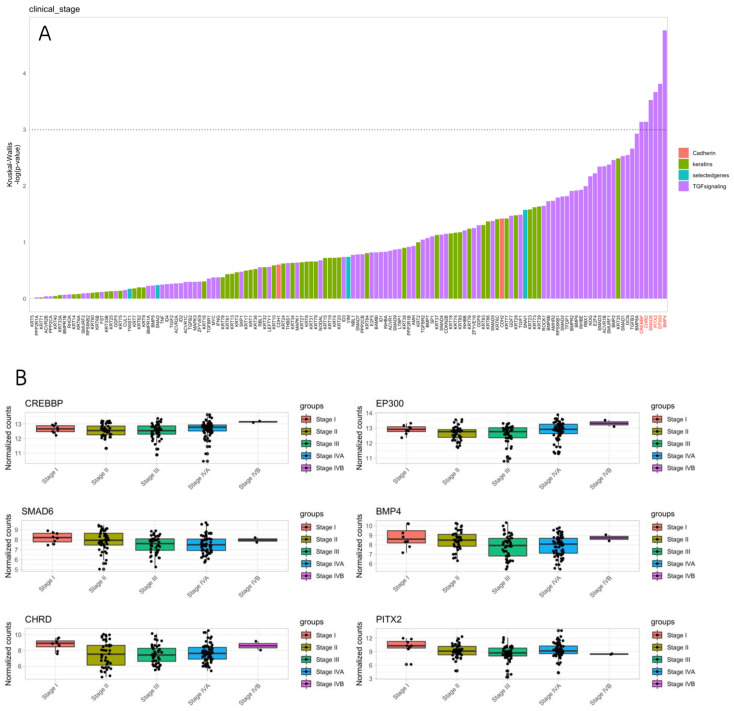
Comparison between TNM staging and genetic analysis. (**A**) Using the Kruskal–Wallis test, the top bar plot illustrates gene expression levels categorized into four groups: red for cadherin genes, green for keratin genes, purple for TGF genes, and blue for selected genes (including VIM, SNAIL1, SNAIL2, TWIST). Genes above the dashed line exhibit a significant difference in expression between the groups (*p*-value < 0.05). (**B**) In the lower box plots, the analysis of differentially expressed genes, as shown in the top bar plot (**A**), is presented in relation to pathological TNM staging. Stage I is represented by the red box, stage II is in the olive green box, stage III is in the green box, stage IVA is in the blue box, and stage IVB is in the purple box. The genes SMAD6, CHRD, and PITX2 were upregulated in early-stage patients, while the EP300 and BMP genes showed significant upregulation in advanced stages.

**Figure 3 cancers-16-01185-f003:**
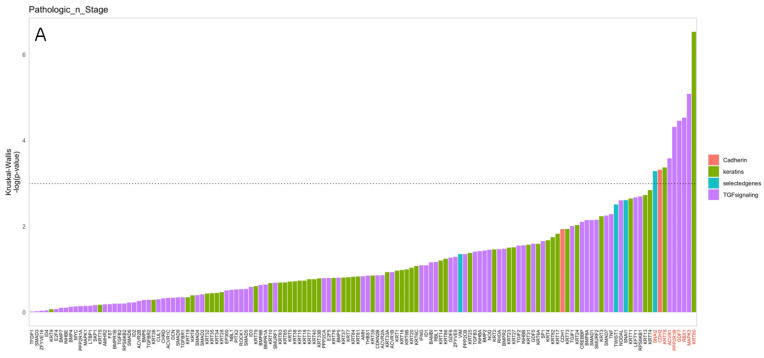
Comparison between cervical lymph node status (N) and genetic analysis. (**A**) The gene expression levels were analyzed using the Kruskal–Wallis test, depicted in the top bar plot. The categories include red for cadherin genes, green for keratin genes, purple for TGF genes, and blue for selected genes (VIM, SNAI1, SNAI2, TWIST). Genes above the dashed line show significant differences in expression between the groups (*p*-value < 0.05). (**B**) In the lower box plots, all studied groups exhibited significant gene expression in relation to the N stage, as shown in (**A**). In the advanced N-stage patients (green and blue boxes), CDH2 (cadherin gene), SNAI2 (selected gene), and ACVR1 gene (TGF signaling) were significantly upregulated. Conversely, the KRT80 gene (keratin group) was upregulated in the N negative group (red box), as shown in the middle lower plot.

**Figure 4 cancers-16-01185-f004:**
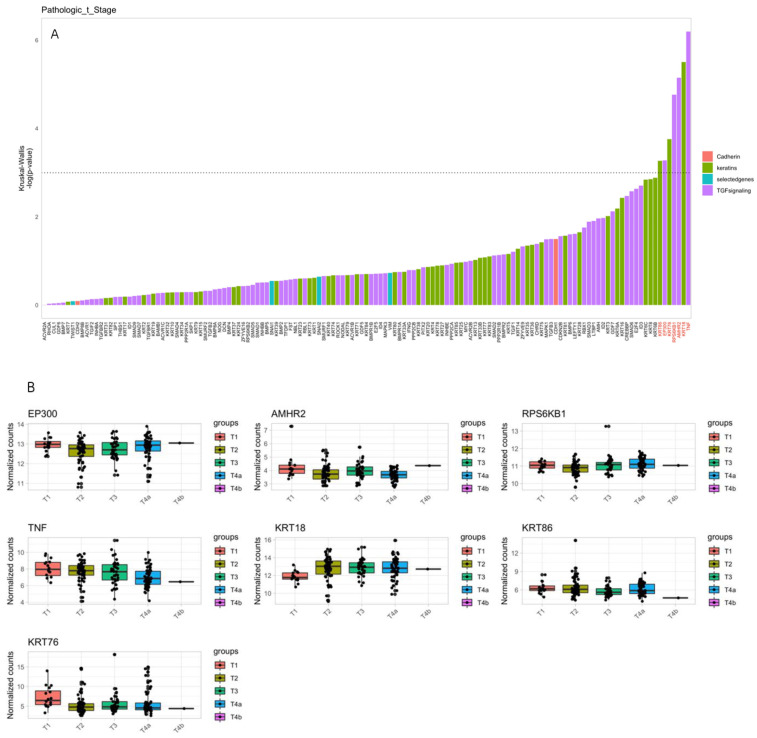
Comparison between tumor size (T stage) and genetic analysis. (**A**) The gene expression levels were analyzed using the Kruskal–Wallis test in the top bar plot. The categories include red for cadherin genes, green for keratin genes, purple for TGF genes, and blue for selected genes (VIM, SNAI1, SNAI2, TWIST). Genes above the dashed line show significant expression differences between the pathological T stages (*p*-value < 0.05). (**B**) In the lower box plots, each significant gene, as documented in (**A**), is analyzed in relation to the different pathological T stages. Keratin genes and TFG genes KRT18, RPS6KB1, and KRT86 were upregulated in the advanced T stage (represented by the green and blue boxes), while KRT76, AMHR2, and EP300 were strongly expressed in the early T stage represented by the red box.

**Figure 5 cancers-16-01185-f005:**
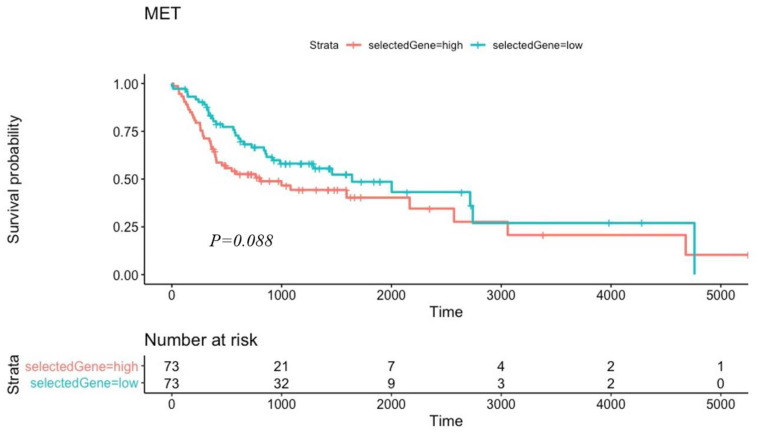
Patients with higher expression of the selected MET genes than the overall mean had lower survival probability than those with lower gene expression.

**Figure 6 cancers-16-01185-f006:**
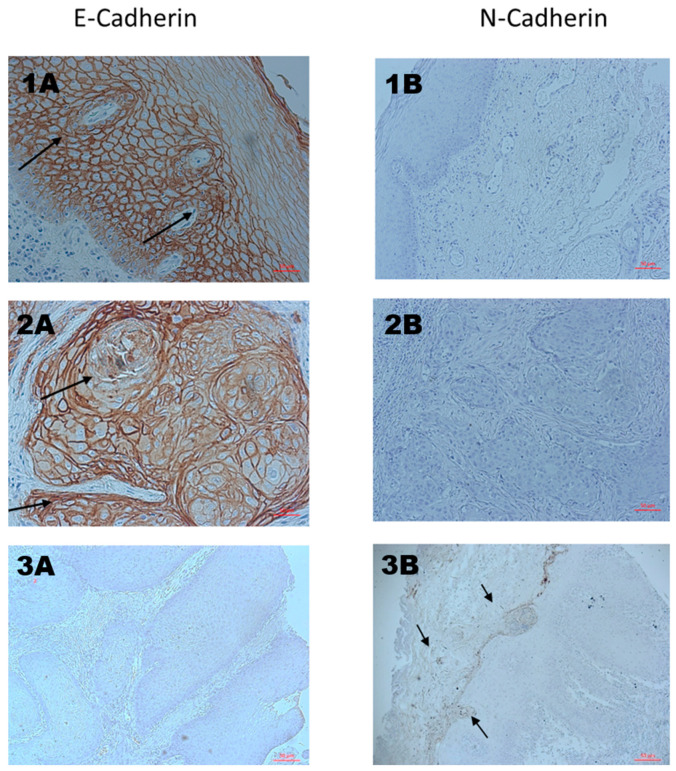
Immunohistochemistry staining of EMT markers in three primary oral squamous cell carcinoma samples (Scale bar 50 μm). The right column shows immunohistochemical staining for E-cadherin, and the left column shows immunohistochemical staining for N-cadherin. **1A** and **2A** display the staining of early-stage samples. Both samples were prominently stained for E-cadherin (indicated by the black arrows, brown staining) and exhibited very weak staining for N-cadherin (**1B** and **2B**). Figure 3 represents an advanced-stage sample, demonstrating strong staining for N-cadherin (**3B**) indicated by black arrows and dark brown staining, and weak staining for E-cadherin (**3A**).

**Table 1 cancers-16-01185-t001:** Clinical and disease-related data of the study cohort.

Characteristic	Study Cohort	“Early Stage” Group	“Advanced Stage” Group	*p* Value
Num.	159	47	112	
Mean age (±STD)	62 ± 13 years	61 ± 17 years	62 ± 12 years	0.3
Male/female	105/54	27/21	78/34	0.06
Tobacco exposure				0.001
Num. (%)	82 (51%)	5 (10%)	77 (69%)	
Av. pack/year	47	16.13	27.91	
Alcohol consumption (%)	101 (63%)	24 (51%)	77 (68%)	0.2
Primary tumor site (%)				0.04
Oral tongue	70 (44%)	24 (51%)	46 (41%)	
Floor of mouth	25 (15%)	4 (8%)	21 (19%)	
Buccal mucosa	8 (4%)	2 (4%)	6 (5%)	
Alveolar ridge	5 (2%)	0	5 (4%)	
Hard palate	4 (2%)	0	4 (3%)	
Lip	1 (0.6%)	0	1 (0.8%)	
Oral cavity *	46 (28%)	17 (36%)	29 (26%)	
p N staging (by H&E)				-
N0 (%)	89 (56%)	47 (100%)	42 (37%)	
N1 (%)	35 (22%)	0	35 (31%)	
N2A (%)	2 (1%)	0	2 (2%)	
N2B (%)	20 (12%)	0	20 (18%)	
N2C (%)	12 (7%)	0	12 (10%)	
N3 (%)	1 (0.6%)	0	1 (0.8%)	
pT staging				-
T1 (%)	9 (5%)	7 (15%)	2 (1.7%)	
T2 (%)	54 (33%)	40 (85%)	14 (13%)	
T3 (%)	45 (28%)	0	45 (40%)	
T4a (%)	50 (31%)	0	50 (44%)	
T4b (%)	1 (0.6%)	0	1 (0.8%)	
TNM staging **				
Stage 1 (%)	7 (4%)	7 (15%)	0	
Stage 2 (%)	40 (25%)	40 (85%)	0	
Stage 3 (%)	41 (26%)	0	41 (36%)	
Stage 4a (%)	69 (43%)	0	69 (61%)	
Stage 4b (%)	2 (1%)	0	2 (2%)	
Surgical margins status				0.06
Negative margins (%)	103 (64%)	35 (74%)	69 (61%)	
Positive margins ^¥^ (%)	19 (12%)	5 (10%)	14 (12%)	
Close margins ^¥^ (%)	37 (23%)	7 (16%)	30 (27%)	
Overall survival (months)	32 months	36 months	19 months	0.01

* Unspecified site in oral cavity. ** Pathological staging according to the American Joint Committee on Cancer (AJCC). ^¥^ For statistical analysis, close margins were considered positive.

**Table 2 cancers-16-01185-t002:** Clinical and disease-related data of the validation cohort.

Characteristic	Study Cohort
Num.	28
Mean age (±STD)	67 ± 13.9 years
Male/Female	16/12
Primary tumor site (%)	
Oral tongue	15
Floor of mouth	2
Buccal mucosa	3
Alveolar ridge	5
Hard palate	0
Lip	3
Oral cavity *	-
p N staging (by H&E)	
N0	13
N1	6
N2A	4
N2B	4
N2C	-
N3	1
pT staging	
T1	14
T2	9
T3	2
T4	3
TNM staging **	
Stage 1	10
Stage 2	6
Stage 3	5
Stage 4	7
IHC staining (ave. %T area)	
E-cadherin (cohort)	23.091 (SD 4.474)
E-cadherin (N negative)	25.919 (SD 4.790)
E-cadherin (N positive)	20.263 (SD 4.582)
N-cadherin (cohort)	0.323 (SD 0.420)
N-cadherin (N negative)	0.171 (SD 0.201)
N-cadherin (N positive)	0.446 (SD 0.442)
Overall survival (months)	33 months

* Oral cavity: subsite not specified. ** 8th Edition of the TNM Oral Cancer staging.

## Data Availability

Data are contained within the article and Appendix A.

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
