# Peer review of "EMT Dynamics in Lymph Node Metastasis of Oral Squamous Cell Carcinoma"

_cancers, 2024, doi:10.3390/cancers16061185_

Round 1

Reviewer 1 Report (New Reviewer)

Comments and Suggestions for Authors

In this manuscript, the authors aimed to evaluate EMT markers as predictors of lymph node metastasis in OSCC. Their results showed that TGF-β/SMAD pathway genes such as SMAD6 were upregulated in advanced stage tumors. N-cadherin and SNAIL2 were overexpressed in node-positive tumors. Keratins were downregulated in these groups. The idea is interesting, but the study was not well designed.

1.   The meaning of the figures was not clearly explained, and whether the expression of these genes in different groups (Fig 2, 3,4) was significantly changed was not analyzed and marked clearly.

2.  The validation cohort contains only 19 samples and was not enough for analysis the expression difference between different groups. Whether the expression of these genes can predict lymph node metastasis of OSCC?

3. The label of Fig 6 was not clear and the authors should detect the expression of these genes in paracancerous tissues, and compare the difference between the samples with or without lymph node metastasis.

Comments on the Quality of English Language

The language should be edited and improved.

Author Response

Dear Reviewer, 

Extensive editing was conducted, new figures and data were added, and the discussion was rewritten. 

please see the attached response to your comments. 

Respond to Reviewer #1:

Thank you for your valuable comments. Please see below response for your comments:

  1. Figures:
  • Figure 2A+B: Comparison Between TNM Staging and Genetic Analysis:

Figure 2A: Using the Kruskal-Wallis test, the top bar plot illustrates gene expression levels categorized into four groups: red for cadherin genes, green for keratin genes, purple for TGF genes, and blue for selected genes (including VIM, SNAI1, SNAI2, TWIST). Genes above the dashed line exhibit a significant difference in expression between the groups (p-value < 0.05).

Figure 2B: In the lower box plots, the analysis of differently expressed genes, as shown in the top bar plot, is presented in relation to pathological TNM staging. Stage I is represented by the red box, stage II is in olive green box, stage III is in the green box, stage Iva is in blue box, and stage IVb is in purple box. The genes "SMAD6," "CHRD," and "PITX2" were upregulated in early-stage patients, while the "EP300" and "BMP" genes showed significant upregulation in advanced stages.

  • Figure 3A+B: Comparison Between Cervical Lymph Node status (N) and Genetic Analysis:

Figure 3A: The gene expression levels were analyzed using the Kruskal-Wallis test, depicted in the top bar plot. The categories include red for cadherin genes, green for keratin genes, purple for TGF genes, and blue for selected genes (VIM, SNAI1, SNAI2, TWIST). Genes above the dashed line show significant differences in expression between the groups (p-value < 0.05).

Figure 3B: In the lower box plots, all studied groups exhibited significant gene expression in relation to the N stage. In the advanced N-stage patients (green and blue boxes), CDH2 (cadherin gene), SNAI2 (selected gene), and ACVR1 gene (TGF signaling) were significantly upregulated. Conversely, the KRT80 gene (keratin group) was upregulated in the N-negative group (red box), as shown in the middle lower plot.

  • Figure 4A+B: Comparison Between Tumor size (T stage) and Genetic Analysis:

Figure 4A: The gene expression levels were analyzed using the Kruskal-Wallis test in the top bar plot. The categories include red for cadherin genes, green for keratin genes, purple for TGF genes, and blue for selected genes (VIM, SNAI1, SNAI2, TWIST). Genes above the dashed line show significant expression differences between the pathological T stages (p-value < 0.05).

Figure 4B: In the lower box plots, each significant gene is analyzed in relation to the different pathological T stages. Keratin genes and TFG genes KRT18, RPS6KB1, and KRT86 were up-regulated in the advanced T stage (represented by the green and blue boxes), while KRT76, AMHR2, and EP300 were strongly expressed in the early T stage represented by the red box.

  1. Validation cohort:

9 more patients were added to the validation study, two patients who were staged as N- negative, and seven patients who were staged as N-positive patients.  Overall, 13 N-negative patients and 15 N-positive patients.

We found that in the N-negative group, E-cadherin was significantly upregulated compared to the N-positive group, and N-cadherin was significantly upregulated in the N-positive group compared to the N-negative group (Student’ t test - P value < 0.05). This finding validates our genetic analysis results in which we found that the E/N cadherin switching pathway plays a significant role in the development of cervical lymph node metastasis, and it could predict the presence of lymph node metastasis.

  1. Figure 6:

The figure was edited and modified.  Advanced-stage tissue staining was added to the figure. The expression of the proteins was also highlighted (lines 329 – 336).

In regard to staining and analyzing para-cancerous tissue, since no precancerous tissue was added in the first part of the study (bio-informatic analysis), we did not include that tissue in our validation study. It may add significant value to the study; unfortunately, since this information is missing from the first and main part of the study, we did not include it.

Reviewer 2 Report (New Reviewer)

Comments and Suggestions for Authors

I think this article is quite complete from the point of view of sampling and genetic analysis, but it is not very robust and has some inconsistencies that I think are serious. The discussion and literature review that has been carried out seems to me to be very inadequate and the objective of applying some of the methodologies, particularly immunohistochemistry, is not well substantiated.

Abstract lines 17-19

In this study, genetic profiling of 159 primary OSCCs from TCGA 15

and analyzed the expression of EMT markers, including cadherin switch genes (CDH1, CDH2), 16

TGF-β/SMAD pathway genes, SNAIL, and keratins Correct this sentence is not readable

Forty-one patients were 176

diagnosed with stage III disease, while the remaining 71 had stage IV disease uniformize numbers

this finding was significantly higher than that in the early-stage 180

group (P value 001). uniformize the appearance of p values

“Karatins genes which included 51 genes . change to keratins

-      Also, it is most important to demonstrate that 231

the genes SNAI2 and CDH2 were remarkably over-expressed in the N2 group, which overexpressed? Uniformize!

-      Figure 3 is to small and impossible to read and analyse, figure 4

-      We found that tumor samples that expressed the MET genes (CDH2 and TGF genes) 260

more than the overall mean expression of those genes had a decreased survival probabil- 261

ity (P value = 0.088). Figure 5  I don´t understand with this p value is important

-      19 primary OSCC 269 this samples is way too small comparing to the 159 patients available for gene expression. Why? What stastistical analysis was use to validate IHQ results?

-      Before the analysis, a senior head and neck 271

pathologist assessed each sample's two H&E stained sections (tail ends) to confirm neo- 272

plastic cellularity. Who? Any of the authors? Why tail ends? Confirm neoplastic cellularity? What that it means?

-          Figure 6: The immunohistochemistry staining images depict the representation of EMT markers in 308

four early-stage oral squamous cell carcinoma samples. Each row displays two sections from the 309

same sample, with the right sections stained for E-cadherin and the left sections stained for N-cad- 310

herin. The images indicate strong staining for E-cadherin, highlighted by arrows in the right column, 311

and weak staining for N-cadherin, as demonstrated in the left column.

Is crucial to improve the quality of this image. What about the later stages?

. SANIL can successfully transfer 357

epithelial cells into a fibroblast-like appearance correct please

-      Several reports have indicated that the extent of 378

EMT markers in ECS could be more indicative of clinical outcomes compared to simply 379

determining the presence or absence of EMT. This suggests that the level or degree of EMT 380

within ECS might hold prognostic and clinical significance [26].  Several reports? Give examples please!

-      Discussion and bibiliography is very poor, the authors didn´t explain enough their own results and the papers pointed out are very few, I think a major change in the discussion and papers cited is needed

Comments on the Quality of English Language

Author Response

Dear Reviewer,

Extensive editing was conducted, new figures and data were added, and the discussion was rewritten. 

Please see the attached response to your comments. 

Respond to Reviewer #2:

Thank you for your valuable comments. Please see below response for your comments:

  1. Abstract lines 17-19:

The sentence was edited and corrected: “ The study involved a genetic profiling analysis of 159 primary OSCCs TCGA samples using bioinformatic tools. The analysis focused on the expression of EMT markers, including cadherin switch genes (N-CDH, E-CDH), TGF-β/SMAD pathway genes, SNAIL, and keratin genes. The samples were categorized into advanced stage (stage III-IV) and early stage (stage I-II).” Lines  15-19.

  1. Forty-one patients were 176 diagnosed with stage III disease, while the remaining 71 had stage IV disease – uniformize numbers

The paragraph was edited, and the numbers uniformed. “ According to the TNM staging (8th edition), the study population was divided into two subgroups Early-stage group, and Advanced-stage group. The advanced-stage group included 112 patients: 41 patients were diagnosed with stage III disease, while the remaining 71 had stage IV disease” lines 174-177.

  1. this finding was significantly higher than that in the early-stage 180 group (P value 001). – uniformize the appearance of p values

The “P value” was edited and uniformed through the entire paper.

  1. “Karatins genes which included 51 genes . change to keratins

Done.

  1. Also, it is most important to demonstrate that 231 the genes SNAI2 and CDH2 were remarkably over-expressed in the N2 group, which – overexpressed? Uniformize!

Done.

  1. Figure 3 is to small and impossible to read and analyse, figure 4

All figures were edited and adjusted.

  1. We found that tumor samples that expressed the MET genes (CDH2 and TGF genes) 260more than the overall mean expression of those genes had a decreased survival probabil- 261ity (P value = 0.088). Figure  I don´t understand with this p value is important

The P value was deleted, and the sentence was edited.

  1. 19 primary OSCC 269 this samples is way too small comparing to the 159 patients available for gene expression. Why? What stastistical analysis was use to validate IHQ results?

 9 more patients were added to the validation study, two patients staged as N- negative, and seven patients staged as N-positive patients.  Overall, 13 N-negative patients and 15 N-positive patients.

We found that in the N-negative group, E-cadherin was significantly upregulated compared to the N-positive group, and N-cadherin was significantly upregulated in the N-positive group compared to the N-negative group (Student’ t-test, P value < 0.05). This finding validates our genetic analysis results in which we found that the E/N cadherin switching pathway plays a significant role in the development of cervical lymph node metastasis, and it could predict the presence of lymph node metastasis.

  1. Before the analysis, a senior head and neck 271 pathologist assessed each sample's two H&E stained sections (tail ends) to confirm neo- 272 plastic cellularity.Who? Any of the authors? Why tail ends? Confirm neoplastic cellularity? What that it means?

  1. Figure 6: Is crucial to improve the quality of this image. What about the later stages?

 The figure was edited and modified.  Advanced-stage tissue staining was added to the figure. The expression of the proteins was also highlighted (lines 329 – 336).

  1. SANIL can successfully transfer 357 epithelial cells into a fibroblast-like appearance correct please

  1. Several reports have indicated that the extent of 378 EMT markers in ECS could be more indicative of clinical outcomes compared to simply 379 determining the presence or absence of EMT. This suggests that the level or degree of EMT 380 within ECS might hold prognostic and clinical significance [26].  Several reports? Give examples please!

  1. Discussion and bibiliography is very poor, the authors didn´t explain enough their own results and the papers pointed out are very few, I think a major change in the discussion and papers cited is needed

Reviewer 3 Report (New Reviewer)

Comments and Suggestions for Authors

357 line SNAIL pg 15 (spelling)

The research design uses 159 TCGA OSCC patients linked to the TCGA database. Rationale and results are provided with adequate rigor and reproducibility to determine significance which is to support the predictive value of selected markers related to growth, metastasis to draining lymph nodes associated with the expression of TGF-beta/E to N-cadherin balance driven is previously reported in human studies data sets explaining lymph node metastasis (Lorenzo-Pouso, etc.  Biology (Basel) Feb 2:13 (2) 239, 2023) The relationship to the genes designated in this report such as SNAIL/SMAD regulating TGF-beta and the other growth regulatory genes is novel particularly effecting lymph node spread for OSCC.

Author Response

Dear reviewer, 

Thank you so much for your valuable comments. 

Please see the attached response to your comments. 

Response to Reviewer #3:

Dear reviewer, thank you so much for your valuable comments.

Please see below respond to your comments:

  1. 357 line SNAILpg 15 (spelling)

Done

Reviewer 4 Report (New Reviewer)

Comments and Suggestions for Authors

The overall paper is well written and presented. However, few points should be stressed out:

1 The equal changes in lymph nodes should be also evaluated in the presence of a well-described control group, whether healthy or another type of cancer.

2 The up-regulation of many of those genes is also seen in the presence of dysbiotic situations, why do not write something about it?

3 The switch from E to N cadherin is not a new topic, the involvement of genes and LN metastasis is also not a new topic...Therefore I would suggest proceeding with a different point of discussion.

4 Some microbes degrade the basement membrane inducing mesenchymal-like cells migration away from the epithelial layer towards other compartments and LN. In addition, microbial intervention causes the downregulation of important epithelial markers like E-cadherin and β-catenin along with upregulation of mesenchymal markers, such as N-cadherin, vimentin, and fibronectin....

Comments on the Quality of English Language

English needs minor revisions

Author Response

Dear Reviewer, 

Extensive editing was conducted, new figures and data were added, and the discussion was rewritten. 

please see the attached response to your comments. 

Respond to Reviewer #4:

Dear Reviewer, Thank you so much for your comments. Please see below a response to your comments:

  1. 1. The equal changes in lymph nodes should be also evaluated in the presence of a well-described control group, whether healthy or another type of cancer.

Unfortunately, we will not be able to address this issue since in the study cohort (the genetic analysis) there was no lymph node specimen available. However, a discussion regarding the genetic analysis of cervical lymph node and EMT was discussed in the discussion section.

Numerous studies have highlighted that disrupted cell-cell adhesion plays a crucial role in promoting cell motility and invasion, which are essential for the establishment of new metastasis. Intriguingly, despite displaying an EMT-like phenotype, LNM specimens consistently exhibited normal E-cadherin levels. This suggests that tumor cells within LNM may employ multiple survival strategies, potentially modulating cell division rates while upholding E-cadherin expression, although displaying characteristics associated with EMT core regulators such as TWIST and SNAIL.

  1. The up-regulation of many of those genes is also seen in the presence of dysbiotic situations, so why not write something about it?

A paragraph regarding this issue was added to the discussion section. (lines 385-413)

3 The switch from E to N cadherin and the involvement of genes in LN metastasis are not new topics. Therefore, I would suggest proceeding with a different point of discussion.

The discussion section was edited, topics such as the E/N cadherin in the lymph nodes, EMT markers and T staging, and the relation between oral microbiota and EMT was added.

4 Some microbes degrade the basement membrane inducing mesenchymal-like cells migration away from the epithelial layer towards other compartments and LN. In addition, microbial intervention causes the downregulation of important epithelial markers like E-cadherin and β-catenin along with upregulation of mesenchymal markers, such as N-cadherin, vimentin, and fibronectin....

A paragraph regarding this issue was added to the discussion section. (lines 385-413).

“Another factors that may induce EMT in OSCC is the Oral Microbiota. The oral cavity harbors a distinctive microbiota comprising over 1000 microbial species, including commensals and opportunistic fungi, bacteria, and viruses. Alterations in this microbial community lead to dysbiosis, exacerbating the onset of oral diseases [37, 38]. Species such as Fusobacterium nucleatum, Porphyromonas gingivalis, and Prevotella sp. have been linked to oral squamous cell carcinoma (OSCC). Prolonged infection of P. gingivalis promotes migratory and invasive properties in epithelial cells, inducing the expression of matrix metalloproteinase and epithelial-mesenchymal transition (EMT) transcription factors like SNAIL, SLUG, and ZEB1. P. gingivalis infection activates signaling cascade pathways such as ERK1/2-Ets1, p38/HSP27, and PAR2/NF-κB to regulate the expression of ProMMP9. Additionally, the gingipain protease produced by P. gingivalis induces the activation of ProMMP9. Phospho-GSK3β, an EMT marker, increases significantly in oral primary epithelial cells during P. gingivalis infection, along with increased expression of EMT transcription factors and decreased expression of adhesion molecules. Long-term infection exacerbates the migration of P. gingivalis-infected epithelial cells and its invasion to neighbor tissue and the lymphatic system [39,40]. In addition, P. gingivalis LPS exacerbates MMP9 secretion by human monocyte-derived dendritic cells, as demonstrated by various dosages inducing MMP9 secretion. Live P. gingivalis strains induce MMP9 and TIMP-1 production, with major and minor fimbriae playing distinct roles in binding and invasion of dendritic cells. Different strains of P. gingivalis influence elevated MMP9 secretion, reflected in upregulated MMP/TIMP-1 ratios. Similarly, Fusobacterium nucleatum increases MMP13 production in epithelial cells, activating MAP kinase p38 and promoting cellular migration through Etk/Bmx, S6 kinase p70, and RhoA kinase. These pathways are known to regulate cytoskeleton organization, focal adhesions, and cellular migration [39]. Fusobacterium nucleatum, similarly to P.gingivalis, can also induce EMT by utilizing adhesins like FadA, Fap2, and outer membrane protein A to bind and invade epithelial cells. FadA specifically binds to E-cadherin, promoting attachment and invasion, ultimately activating β-catenin signaling and leading to the upregulation of Wnt genes, transcription factors, N-cadherin and other oncogenes like myc and cyclin D1. This process involves phosphorylation and internalization of E-cadherin, affecting the phosphorylation and nuclear translocation of β-catenin, which eventually leads to degration of the basement membrane, and migration of the epithelial cells [41]. “

Round 2

Reviewer 1 Report (New Reviewer)

Comments and Suggestions for Authors

The concerns were well addressed by the authors and the manuscript was significantly improved.

Reviewer 2 Report (New Reviewer)

Comments and Suggestions for Authors

The authors adressed the main issues detected in the revision, and the paper  has been through a great improvement and it is know more complete and conscise.

Before proceeding with the Immunohisto-289 chemical analysis, an independent head and neck pathologist reviewed two sections of 290 each sample(including tissue margins) to verify the presence of oral squamous cell carcinom (the letter "a" is missing here)

Reviewer 4 Report (New Reviewer)

Comments and Suggestions for Authors

The manuscript can be accepted

This manuscript is a resubmission of an earlier submission. The following is a list of the peer review reports and author responses from that submission.

Round 1

Reviewer 1 Report

Comments and Suggestions for Authors

The authors of the manuscript emphasize a significant correlation between the expression of epithelial-mesenchymal transition (EMT) markers and lymph node metastasis in oral squamous cell carcinoma (OSCC). Despite the extensive body of research in this field, the manuscript appears to lack the necessary novelty regarding the exploration of EMT in OSCC. Moreover, there is a notable absence of discussion regarding extracapsular spreading (ECS) in the manuscript. It remains unclear whether the authors have addressed this crucial aspect in their research. Further elucidation on ECS could enhance the comprehensiveness of the manuscript and contribute valuable insights to the understanding of OSCC progression.

Author Response

Dear Reviewer, thank you for your valuable insight.

Unfortunately, we only had 2 cases of ECS. Thus, we could not conclude the specific expression of EMT markers on positive ECS samples.

However, a paragraph regarding the importance of EMT in ECS was added to the discussion section.

“The relationship between Extra Capsular Spread (ECS) and Epithelial-Mesenchymal Transition (EMT) is not clearly understood. Several reports have indicated that the extent of EMT markers in ECS could be more indicative of clinical outcomes compared to simply determining the presence or absence of EMT. This suggests that the level or degree of EMT within ECS might hold prognostic and clinical significance [45].

Some research findings have suggested that EMT may play a significant role in the development of ECS. For example, certain studies have found a notable presence of an EMT expression signature specifically at the forefront of tumor margins, implying that EMT could be associated with the invasive and migratory characteristics of cancer cells, which are necessary for developing ECS [46]. Furthermore, the presence of EMT markers in positive lymph nodes expressing ECS has been linked to poorer clinical outcomes; patients with ECS-positive lymph nodes and concurrent EMT marker expression exhibited a five times worse survival rate compared to patients without EMT marker expression [47].

These findings underscore EMT’s potential significance in the context of ECS and cancer progression. Understanding the intricate relationship between ECS and EMT is crucial for developing more effective prognostic markers and targeted therapies to address aggressive cancer behavior associated with ECS.”

Reviewer 2 Report

Comments and Suggestions for Authors

Authors made great efforts in paving the importance of EMT in determining the prognosis of oral cavity cancers.

Here are few comments and suggestions:

1. Line 89-93 -  inclusion criteria was hpv negative OSCC (243) then what excluded 84 patients from the final analysis as only 159 patients were included in final analysis ?

2. Line 94- 96 - it would be better to use universal terms "Early stage and advanced stage " instead of low and high grade for stage 1&2 and stage 3& 4 respectively. It should be changed overall in the manuscript.

3. Line 142- Table 1 shows it to be 50% 

4. Table 1 - Advanced stage tumors had higher incidence of tobacco consumption as compared to early stage- this can be a confounder for nodal metastasis or  proponent for EMT - Authors need to comment on it as in develeoping countries - majority of thr oral cavity cancers (oscc) are tobacco users.

5. Table 1 has few fallicies or its less understood 

pN stage,pT stage, Overall stage dont add up to 100% - it needs to be modified 

6. Studies have shown that EMT also have higher chances of bone invasion and other risk factors which may influence overall survival.

7.  Line 162 - While recurrence is certainly a crucial factor influencing survival, it's not the sole determinant, 

as 30% of the recurrent and second primary are treatable tumors with survival thus this statement requires your attention.

8. Line 235 - Authors need to correct the statement as p value is more than 0.05 therefore it cannot be termed as statistically significant though it had trends towards it

9. Line 238 - Language needs to be corrected as p value is more than 0.05 .

10. Line 255 - 257 - Did author assess the p value of the same ?

Comments on the Quality of English Language

no

Author Response

Thank you for your valuable comments.

Please see below the response to your comments:

  1. Line 89-93 -  inclusion criteria was hpv negative OSCC (243) then what excluded 84 patients from the final analysis as only 159 patients were included in final analysis ?

Out of the 243 HPV(-) patients, only 159 patients had Oral Squamous Cell Carcinoma (72 cases of Larynx, 2 cases of hypopharynx, and 10 cases of Tonsils).  This was added to the text lines: 89-91.

  1. Line 94- 96 - it would be better to use universal terms "Early stage and advanced stage " instead of low and high grade for stage 1&2 and stage 3& 4 respectively. It should be changed overall in the manuscript.

Done

  1. Line 142- Table 1 shows it to be 50% 

The numbers and ratios were reviewed and corrected, and table 1 was updated.

  1. Table 1 - Advanced stage tumors had higher incidence of tobacco consumption as compared to early stage- this can be a confounder for nodal metastasis or  proponent for EMT - Authors need to comment on it as in develeoping countries - majority of thr oral cavity cancers (oscc) are tobacco users.

A paragraph regarding this topic was added to the discussion.

Another interesting finding in the current study was that patients in the advanced group reported a significantly higher tobacco consumption rate compared to the early-stage group. Several studies demonstrated that nicotine significantly impacts tumor cells' epithelial-mesenchymal transition (EMT) [47]. Nicotine is the primary component of tobacco, can induce oxidative stress in tumor cells. This oxidative stress can trigger signaling pathways that promote EMT. As a result, tumor cells treated with nicotine demonstrate increased expression of mesenchymal markers such as Vimentin, Snail, and PrX1 proteins while showing decreased levels of epithelial markers like E-cadherin [48-50]. The upregulation of mesenchymal markers and downregulation of epithelial markers in response to nicotine exposure suggest that nicotine may play a role in promoting the transition of tumor cells toward a more aggressive, invasive phenotype. This transition is a hallmark of EMT and is associated with enhanced metastatic potential and resistance to cancer treatments [48].In addition to its role in inducing EMT, nicotine has been linked to other aspects of cancer progression, including the promotion of tumor growth, angiogenesis, and resistance to cell death. Further research in this area may provide insights into potential strategies for preventing or reversing the effects of nicotine on EMT and its associated consequences in cancer.

  1. Table 1 has few fallicies or its less understood: pN stage,pT stage, Overall stage dont add up to 100% - it needs to be modified 

Done

  1. Studies have shown that EMT also have higher chances of bone invasion and other risk factors which may influence overall survival.

Lines: 419-431

EMT was also found to be involved in bone invasion and progression. The precise molecular processes involved in bone invasion in oral squamous cell carcinoma (OSCC) are not thoroughly comprehended. However, heightened levels of cytokines and growth factors are observed within the extracellular matrix, leading to the recruitment of osteoclasts and subsequent bone resorption. This sets the stage for neoplastic cells to establish a conducive environment within the bone, promoting their growth and division, thereby initiating and perpetuating a destructive cycle. Moreover, this microenvironment has the potential to influence neoplastic cell behavior, potentially triggering epithelial-mesenchymal transition (EMT), which in turn sustains this deleterious cycle. Studies have shown that EMT markers were expressed in the tumor-bone interface surface, which may indicate the role of those markers in inducing bone invasion. Twist, a well-known factor in the process of EMT, was found to be positively expressed in the interaction between OSCC and bone tissue in the infiltrative invasion pattern and with the absence of periostin [51-53].

  1. Line 162 - While recurrence is certainly a crucial factor influencing survival, it's not the sole determinant, as 30% of the recurrent and second primary are treatable tumors with survival thus this statement requires your attention.

Although recurrence is not the only factor determining the survival of OSCC pa-tients, in this study the patients’ survival status was studied as a function of the recur-rence status at the last follow-up visit. Disease free survival was significantly lower among patients in the advanced-stage group (mean: 19 months versus 36 months, P value =.01).

  1. Line 235 - Authors need to correct the statement as p value is more than 0.05 therefore it cannot be termed as statistically significant though it had trends towards it

To maximize our genetic analysis and to decrease the false positive ratio, we sat the genetic analysis ratio to be significant at Deseq cutoff at P<0.1.

  1. Line 238 - Language needs to be corrected as p value is more than 0.05 .

To maximize our genetic analysis and to decrease the false positive ratio, we sat the genetic analysis ratio to be significant at Deseq cutoff at P<0.1.

  1. Line 255 - 257 - Did author assess the p value of the same?

Yes

Reviewer 3 Report

Comments and Suggestions for Authors

The authors described "EMT Dynamics in Lymph Node Metastasis of Oral Squamous Cell Carcinoma" using genetic profiling of 159 primary OSCCs from TCGA and analyzed the expression of EMT markers, including cadherin switch genes (CDH1, CDH2), TGF-β/SMAD pathway genes. They concluded that these genes are responsible for the EMT progression, and their expression enables tumor cells to migrate and invade the cervical lymph nodes. These results should be attractive and informative for potential readers. I have some questions and recommendations to more improve this manuscript.

1. Figures 4 and 5 are different sizes, please match the sizes. 4 is hard to see. Also, there are some confusing words. "low stage" or "Low stage"?

2. In Figure 6, please indicate EMT marker using arrow heads or arrows. In addition, please make it a larger image. Why are there 8 images? It is hard to understand. More figure legends should be added.

3. How about EMT marker in lymph nodes? Please add it.

Comments on the Quality of English Language

None.

Author Response

Thank you so much for your valuable comments.

Please see below a respond.

  1. Figures 4 and 5 are different sizes, please match the sizes. 4 is hard to see. Also, there are some confusing words. "low stage" or "Low stage"?

Done

  1. In Figure 6, please indicate EMT marker using arrow heads or arrows. In addition, please make it a larger image. Why are there 8 images? It is hard to understand. More figure legends should be added.

Arrows were added to the figures. In this figure, 8 different samples, all negative for lymph node metastasis, are presented. In the staining, we can see low expression for N-cadherin and high staining for E-cadherin.

“Immunohistochemistry staining representation of EMT markers in oral tumor sections. Staining shows high expression of E-cadherin compared with N-cadherin in N-negative patients in 8 different samples.”

  1. How about EMT marker in lymph nodes? Please add it.

Lines 500-514:

In the context of lymph node metastasis (LNM), several studies have found a notable correlation between reduced E-cadherin in the primary tumor cells and the involvement of multiple lymph nodes [54], suggesting a potential association with enhanced tumor dissemination within regional lymphatics. However, a shift from negative to positive E-cadherin status was found in the LNM cells [54,55,56], which demonstrates an association between E-cadherin loss and lymph node involvement [55,56] as well as the re-expression of E-cadherin in metastases. Numerous studies have highlighted that disrupted cell-cell adhesion plays a crucial role in promoting cell motility and invasion, which are essential for the establishment of new metastatic sites. In contrast, the reversion to an epithelial state at a distant site may be vital for tumor cell survival. Intriguingly, despite displaying an EMT-like phenotype, LNM specimens consistently exhibited normal E-cadherin levels. This suggests that tumor cells within LNM may employ multiple survival strategies, potentially modulating cell division rates while upholding E-cadherin expression, although displaying characteristics associated with EMT core regulators such as TWIST1, SNAI1, or SNAI2 [55-58].

Round 2

Reviewer 2 Report

Comments and Suggestions for Authors

Majority of the comments and suggestion are well integrated .

In regards to the 

  1. Line 235 - Authors need to correct the statement as p value is more than 0.05 therefore it cannot be termed as statistically significant though it had trends towards it

To maximize our genetic analysis and to decrease the false positive ratio, we sat the genetic analysis ratio to be significant at Deseq cutoff at P<0.1.

  1. Line 238 - Language needs to be corrected as p value is more than 0.05 .

To maximize our genetic analysis and to decrease the false positive ratio, we sat the genetic analysis ratio to be significant at Deseq cutoff at P<0.1.

The above exercise is an uncommon practise, I understand to screen the factors for its association , it may work but it should not be used in standard practise,. As it will be difficult to derieve conclusion based on these results 

Reviewer 3 Report

Comments and Suggestions for Authors

The authors revised the manuscript. However, there are a lot of immature points.

1. Why did you change co-authors at this stage? Usually, co-authorship should be discussed either during the preparation of the paper or when the paper is first written.

2. In Tabel 1, is this mean age standard deviasion right? I cannot believe this number.

3. Discussion part was too long and superfluous.

4. Although I pointed out in Figure 6 legends, they did not answer my qustion.

There are many points of skepticism.

Comments on the Quality of English Language

I leave it to editor.